# The Effect of Natural-Based Formulation (NBF) on the Response of RAW264.7 Macrophages to LPS as an In Vitro Model of Inflammation

**DOI:** 10.3390/jof8030321

**Published:** 2022-03-21

**Authors:** Sheelu Monga, Basem Fares, Rami Yashaev, Dov Melamed, Meygal Kahana, Fuad Fares, Abraham Weizman, Moshe Gavish

**Affiliations:** 1Department of Human Biology, Faculty of Natural Sciences, University of Haifa, Haifa 3498838, Israel; sheelumonga@hotmail.com (S.M.); ffares@univ.haifa.ac.il (F.F.); 2Ruth and Bruce Rappaport Faculty of Medicine, Technion-Israel Institute of Technology, Haifa 3200003, Israel; rami.yashaev@campus.technion.ac.il (R.Y.); meygal49@gmail.com (M.K.); 3Cannabotech Ltd., 3 Arik Einstein St., Herzliya 4659071, Israel; basem@cannabotech.com (B.F.); dm@cannabotech.com (D.M.); 4Sackler Faculty of Medicine, Felsenstein Medical Research Center, Tel Aviv University, Tel Aviv 6997801, Israel; weizmana@gmail.com; 5Research Unit, Geha Mental Health Center, Petah Tikva 4910002, Israel

**Keywords:** inflammation, cytokines, chemokines, macrophages, medicinal mushrooms, lipopolysaccharide, cannabidiol

## Abstract

Macrophages are some of the most important immune cells in the organism and are responsible for creating an inflammatory immune response in order to inhibit the passage of microscopic foreign bodies into the blood stream. Sometimes, their activation can be responsible for chronic inflammatory diseases such as asthma, tuberculosis, hepatitis, sinusitis, inflammatory bowel disease, and viral infections. Prolonged inflammation can damage the organs or may lead to death in serious conditions. In the present study, RAW264.7 macrophages were exposed to lipopolysaccharide (LPS; 20 ng/mL) and simultaneously treated with 20 µg/mL of natural-based formulation (NBF), mushroom–cannabidiol extract). Pro-inflammatory cytokines, chemokines, and other inflammatory markers were analyzed. The elevations in the presence of interleukin-6 (IL-6), cycloxygenase-2 (COX-2), C-C motif ligand-5 (CCL5), and nitrite response, following exposure to LPS, were completely inhibited by NBF administration. IL-1β and tumor necrosis factor alpha (TNF-α) release were inhibited by 3.9-fold and 1.5-fold, respectively. No toxic effect of NBF, as assessed by lactate dehydrogenase (LDH) release, was observed. Treatment of the cells with NBF significantly increased the mRNA levels of TLR2, and TLR4, but not NF-κB. Thus, it appears that the NBF possesses anti-inflammatory and immunomodulatory effects which can attenuate the release of pro-inflammatory markers. NBF may be a candidate for the treatment of acute and chronic inflammatory diseases and deserves further investigation.

## 1. Introduction

Macrophages play an important role in the host-immune interaction during infection [1]. Their activation may lead to various acute and chronic inflammatory diseases such as rheumatoid arthritis, tuberculosis, asthma, sinusitis, Crohn’s disease, ulcerative colitis, hepatitis, and viral infections [2,3,4]. Macrophages can be activated by different stimuli, including lipopolysaccharide (LPS, an endotoxin from Gram-negative bacteria), leading to the release of pro-inflammatory cytokines such as tumor necrosis factor-alpha (TNF-α), interleukin (IL)-1β and IL-6 [5], inflammatory mediators such as nitric oxide (NO), prostaglandin (PG) E2, and the production of reactive oxygen species (ROS), which significantly contribute to the induction of peripheral and neuro-inflammatory diseases [6,7]. Medicinal mushrooms are a rich source of free amino acids, carbohydrates, proteins, vitamins, and various essential minerals and trace elements [8,9]. In addition, medicinal mushrooms include bioactive metabolites that hold a high medicinal value such as lectins, polysaccharides, phenolics and polyphenolics, terpenoids, ergosterols, and volatile organic compounds [10,11,12]. Mushrooms are well known as biological response modifiers (BRM) adaptogens, or immune activators, which are capable of stimulating immune functions.

In the present study, five medicinal mushrooms and cannabidiol (CBD) were used in a single formulation designated as a natural-based formulation (NBF).

The NBF contained five mushrooms as follows: *Trametes versicolor* (Turkey Tail mushroom), with proven immunomodulating properties due to two unique polysaccharide–protein complexes, that are soluble in water [13], and have been used clinically in Japan, China, and Korea as an adjuvant therapy for a variety of cancers for more than 30 years. PSK increased the concentrations of lymphocytes, macrophages, and neutrophils in bronchoalveolar lavage fluid and induced the synthesis of TNF-α, IL-1α, IL-6, MIP-1α, and MIP-1β31 in patients, and improved both their survival and disease-free rates [14]; *Ganoderma lucidum* (Lingzhi or Reishi mushroom), with proven immunomodulating properties due to its combination of high polysaccharide contents (up to 40% of beta-glucans) and triterpenoid compounds (approximately 150 different kinds of triterpenoids). They are classified into several groups based on their carbon numbers and states of oxidation (ganoderic, ganoderemic and ganolucidic acids). The addition of alcohol extract of *Ganoderma lucidum* to the formulation increased the quantity of triterpenoids in the preparation [15,16]. These extracts modulate the NF-κB/MMP-9 pathways through blocking the process of phosphorylation and degradation of IκBα, resulting in suppression of the nuclear translocation of p65. Furthermore, they attenuate the production of pro-inflammatory cytokines, including IL-6 and IL-1β, in LPS-induced RAW264.7 cells [17]; *Lentunus edodes* (Shiitake mushroom), with proven immunomodulating properties due to lentinan, a highly purified polysaccharide that contains glycoproteins, ergosterol, and other compounds. Lentinan extracts reduce cytokine-induced NF-κB activation in A549 epithelial cells and decrease the levels of pro-inflammatory cytokine production (TNF-α, IL-8, IL-2, IL-6, IL-22) [18]; *Flammulina velutipes* (Velvet Foot mushroom), with proven immunomodulating properties related to the presence of EA6, a protein-bound polysaccharide, proflammin (90% protein and 10% polysaccharide) and protein Fve [19,20]; and *Cordyceps militaris* (Caterpillar mushroom), with proven immunomodulating properties due to specific polysaccharides, lipids, and a large number of nucleoside analogues, the most important of which is cordycepin (3-deoxyadenosine), a derivative of adenosine [19,21]. Cordyceps extracts decrease the NO production and the differentiation of macrophages and increase the proliferation of RAW264.7 macrophages [22].

These five mushrooms activate an immunosuppressed system via stimulation of innate and adaptive immunity, by the binding of specific ligands to the pathogen recognition receptors (PRRs) which initiate both innate and adaptive immunological responses to specific pathogens such as fungi, bacteria and yeast membranes [23]. Binding to membrane-bound PRRs that include Toll-like receptors (TLRs), C-type lectin receptors (CLRs), dectin-1 and complement receptor 3 (CR3) by pathogen-associated molecular patterns (PAMPs) such as polysaccharides can activate immune responses by enhancing the secretion of TNF-alpha, IL-6 and other inflammatory cytokines [24]. NF-κB receives input from a different type of receptor and inflammatory inducer such as TNF-α to promote specific cellular responses that are suitable to the stimulating inducer [25]. Pathogen-recognition receptors also serve to bind ligands that prime immune responses [26,27].

CBD is a plant-derived cannabinoid extracted from cannabis and comprises a terpene and an aromatic ring (1); it affects the endocannabinoid system by acting as a CB2 agonist and affecting various immune functions [28,29].

Mushroom extracts (ME) and CBD were combined in an attempt to achieve augmented immunomodulatory activity [30,31]. TLR4 triggers signaling cascades following LPS binding (e.g., NF-κB and MAPK) that leads to the production of proinflammatory cytokines (such as TNF-α, IL-1β, IL-6, IL-12) and type-I interferons that are required for the pro-inflammatory response and elimination of pathogens [32].

Macrophages play different roles in maintaining homeostasis and a normal physiological condition by adjusting various biological activities [33]. LPS is widely used as an inducer of inflammatory responses [34]. In the present study, the in vitro immunomodulatory effect of NBF (mushroom–cannabidiol extract, Cannabotech, Herzliya, Israel) was assessed in murine RAW264.7 macrophage inflammatory responses to LPS.

It was previously shown that ME and CBD can attenuate the inflammatory responses of macrophages to LPS stimulation [29,30,31,32,33,34,35]. Based on these observations, we hypothesized that NBF, which contains a combination of both botanical components, can serve as a potent inhibitor of over-production of inflammatory mediators by macrophages exposed to LPS. Such a combined extract may be a candidate for the prevention and suppression of acute cytokines storm and inflammatory responses in the periphery and the brain. 

## 2. Materials and Methods

### 2.1. RAW264.7 Murine Macrophage Cells

RAW264.7 macrophage cells (provided as a kind gift by Prof. Tsaffrir Zor, Tel Aviv University, Tel Aviv, Israel) were cultured at 37 °C with 5% CO_2_ and 90% relative humidity. Cells were incubated in Dulbecco’s modified Eagle’s medium (DMEM) containing 4.5 g/L glucose, 2 mM L-glutamine, and supplemented with 10% fetal bovine serum (FBS), 1% sodium pyruvate, penicillin (100 U/mL), and streptomycin (100 µg/mL) (all the culture materials were purchased from Biological Industries, Beit Ha’Emek, Israel).

### 2.2. Preparation of NBF

The NBF contained 80% of the hot water extracts of equal portions of five medicinal mushrooms (Trametes versicolor (Turkey Tail mushroom), Lentunus edodes, (Shiitake mushroom), Flammulina velutipes (Velvet Foot mushroom), Cordyceps militaris (Caterpillar mushroom) and 10% of alcoholic extract of Ganoderma lucidum (Lingzhi or Reishi mushroom)), 10% Ganoderma lucidum alcoholic extract, and 10% isolated cannabidiol (CBD) > 98%, (CAS# 13956-29-1) which was purchased from Mile High Labs, (Loveland, CO, USA). All components of the NBF were provided by Cannabotech Co. Herzliya, Israel, and dissolved in vehicle (30% DMSO + 0.5% Tween20). The active substances in the ME are summarized in Table 1. 

The ME were prepared according to the method described before (Huang et al., 2021) [36]. Briefly, mixtures of mycelial powders were used for the extraction of bioactive compounds; distilled water was used for hot water extraction; dry powder of mycelial biomass was extracted for 3 h with distilled H_2_O (1 g/10 mL proportion) at 80 °C using (Shaking Water Bath, (MRC, Holon, Israel). Alcoholic extracts were prepared from chipped fruiting bodies of Ganoderma lucidum (1 kg) with ethanol (95%, 1 L) for 24 h at room temperature, to reach 35 g of solid extract. After extraction, insoluble solid compounds were separated by centrifugation at 6000 rpm at 4 °C for 15 min followed by filtration through the Whatman filter paper N 1. Filtrates were evaporated using a vacuum evaporator and finally dried at 40 °C in an air-forced laboratory oven [36]. 

### 2.3. Exposure of RAW264.7 Cells to LPS 

RAW264.7 cells were seeded in a 12-well plate (10^5^ cells per well) in serum containing medium and grown for 24 h. The cells were exposed to 20 ng/mL of LPS (Sigma, St. Louis, MO, USA) for the next 24 h.

### 2.4. Exposure of RAW264.7 Cells to NBF

Five medicinal mushrooms (MMs) were used in the present study: *Trametes versicolor*, *Cordyceps militaris*, *Flammulina velutipes*, *Ganoderma lucidum*, and *Lentinus edodes*. Pure cultures of tested alcohol extracts from MM species along with isolated hemp-derived CBD were obtained from Cannabotech (Herzliya, Israel). Extracts were dissolved and combined to form a homogenous mix and stored at 4 °C.

RAW264.7 cells were treated with 20 µg/mL of NBF in all experiments with or without exposure to LPS. This concentration of NBF was chosen since it was the minimal most effective concentration shown in dose–response analysis (Appendix A). The cell culture supernatant was collected, aliquoted, and stored at −20 °C until use.

### 2.5. Trypan Blue Staining

The cells were scrapped off before seeding into a 12-well plate. This procedure included 1:1 ratio of cells and trypan blue dye in an Eppendorf tube. Ten microliters of this mixture was loaded into a Neubauer’s slide covered with a slip. A total of 10^5^ cells were seeded per well in the 12-well plate in complete medium.

### 2.6. Enzyme Linked Immunosorbent Assay (ELISA)

Sample preparation: Supernatants of cell culture were carefully collected from the 12-well plate, aliquoted, and stored at −20 °C until they were assayed.

Plate preparation and standards for IL-1β (ab197742), IL-6 (ab222503), TNF-α (ab208348), and COX-2 (ab210574) (all the kits were purchased from Abcam, Zotal Ltd., Tel Aviv, Israel) ELISA kits were provided as ready to use according to the manufacturer’s instructions. Standards were serially diluted with a diluent and the samples were defrosted at room temperature. Antibody cocktail (capture and detection antibody) was diluted in antibody diluent according to the manufacturer’s instructions. A 50 µL mixture of standards and the prepared antibody was added to the wells. The wells were sealed and incubated for 1 h at room temperature on a plate shaker at 100 rotations per minute (rpm). Each plate was washed 4 times with 300 µL of washed buffer. TMB substrate (100 µL) was added, and samples were incubated for 5–15 min in the dark on a plate shaker (depending upon the appearance of color). To stop the reaction, 100 µL of stop solution was added and mixed on a shaker for 1 min. Optical density (O.D.) was recorded at 450 nm with endpoint reading [7,36].

For CCL5 (ab213736), an NUNC 96-well plate was coated with 50 µL of 2 µg/mL capture antibody (diluted in coating buffer) for 2 h at room temperature on a shaker (100 rpm). Then, the plate was washed 4 times with 300 µL of wash buffer and blocked with 300 µL blocking buffer at 4 °C on a shaker overnight. The next day, the plate was washed again 4 times in 300 µL of wash buffer and 50 µL of standard and sample was used. The plate was again incubated for 2 h at room temperature on a shaker. The washing process was repeated, and 50 µL of 0.5 µg/mL of detection antibody was incubated for 2 h at room temperature on a shaker. The plate was washed after 2 h, and 0.05 µg/mL streptavidin was added (50 µL in each well) and incubated at room temperature for 1 h on a shaker. The plate was washed as described before and 100 µL of TMB substrate was used in the plate and incubated for 5–15 min until colorization appeared. Stop solution was used to stop the reaction and the O.D. was recorded at 450 nm with endpoint reading.

### 2.7. Nitrite Assay

RAW264.7 cells were seeded in a 12-well plate and incubated for 24 h in complete medium. Then, the cells were exposed to 20 ng/mL of LPS and treated with NBF (20 µg/mL). Nitrite production was determined by measuring the levels of the NO metabolite nitrite (NO_2_) in the medium using a colorimetric reaction with Griess reagent. Then, 100 µL of cell culture supernatant and Griess reagent was mixed (1:1). Sodium nitrate, 0.1 M, was used to build a calibration curve. Absorbance was measured at 540 nm after 15 min on a shaker (50 rpm), with a Spectrophotometer Zenyth 200 (Anthos, Eugendorf, Austria) [7].

### 2.8. LDH Assay

When plasma membranes of the cell undergo damage, cytoplasmic enzyme LDH is released as an indicator of necrosis. Cytotoxicity was analyzed using a lactate dehydrogenase cytotoxicity assay kit (ab102506, Cambridge, MA, USA). The cells were seeded in a 96-well plate (10,000 cells per well) and incubated for 24 h. The cells were exposed to LPS and treated with NBF in low-serum medium (1% FBS). Then, 100 µL of cell culture supernatant was transferred into a new plate. The reaction mixture from the kit was added to the sample in a new plate (1:1) and incubated for 15 min on a shaker at 50 rpm. The formed amount of formazan was proportional to the amount of LDH release. The color intensity was proportional to the number of damaged cells. The O.D. was read at 492 nm with reference wavelength of 690 nm at endpoint reading with Spectrophotometer Zenyth 200 (Anthos, Eugendorf, Austria).

### 2.9. Real-Time PCR

RNA was extracted from lysed cells using an RNeasy™ Mini Kit (Qiagen, Hilden, Germany). Reverse transcription was performed using a high-capacity cDNA reverse transcription kit (ThermoFisher Scientific, Waltham, MA, USA). Real-time qPCR was performed using Fast SYBR Green (ThermoFisher Scientific, Waltham, MA, USA), and analyzed in a StepOnePlus™ PCR machine (for the list of primers used, see Appendix A). All experiments were performed in three independent replicates where the average and standard error of the mean of all three were calculated. *p*-value was calculated by Student’s *t*-test.

### 2.10. Statistical Analyses

Results are presented as the mean ± standard deviation (SD) or standard error of the mean (SEM). One-way analysis of variance (ANOVA) tests were used as appropriate, followed by Bonferroni’s post hoc test. Statistical significance was defined as *p* < 0.05. In some of the experiments, Student’s *t*-test was performed in comparison to control (LPS alone).

## 3. Results

### 3.1. Cytotoxicity Analysis with LDH Assay

In the first stage of the project, we assessed a possible cytotoxic effect of NBF. To this end, we measured the LDH release from RAW264.7 macrophages exposed to NBF. The LDH release was compared between cells exposed to NBF and cells exposed to its corresponding vehicle, compared to naïve cells. No toxic effect was detected in cells exposed to NBF compared to controls (Figure 1).

### 3.2. Pro-Inflammatory Cytokines

As shown in Figure 2, LPS induced a dramatic increase in the inflammatory mediators IL-1β (A), IL-6 (B), and TNF-α (C), while vehicle did not significantly affect the levels of these mediators. Naïve cells and vehicles did not affect the release of the inflammatory mediators. Exposure of the cells to LPS + vehicle resulted in a significant increase in the inflammatory mediators. However, exposure of the cells to LPS + NBF significantly counteracted the release of pro-inflammatory cytokines. The release of IL-6 was completely inhibited (Figure 2B). The release of IL-1β was inhibited by 3.9-fold as compared to LPS alone group (Figure 2A), whereas TNF-α was inhibited by 1.5-fold (Figure 2C). NBF did not show any effect on its own.

### 3.3. Chemokine

The effect of NBF on CCL5 (RANTES) release was measured, as a part of the pro-inflammatory response using ELISA. Naïve and vehicle did not affect CCL5 release, while CCL5 levels were significantly increased in the LPS and LPS + vehicle groups as compared to vehicle. The release of CCL5 was completely inhibited following exposure of the cells to LPS + NBF. NBF alone did not affect CCL5 levels (Figure 3).

### 3.4. Intracellular Inflammatory Marker-COX-2

COX-2 levels were measured by ELISA using cell lysate samples as a part of intracellular inflammatory response. LPS increased the level of COX-2 protein up to 5.3-fold compared to naïve/vehicle groups and LPS+ vehicle have also shown a similar effect. In contrast, in the LPS + NBF group, the release of COX-2 was almost completely suppressed (insignificant compared to naïve/vehicle groups). NBF alone did not affect COX-2 release (Figure 4).

### 3.5. Nitrite Assay

Nitrite levels in the RAW264.7 macrophage cell culture supernatant were assessed by Griess reaction. The nitrite levels were increased up to 56.9-fold following exposure to LPS in comparison to naïve/vehicle groups. LPS+ vehicle did not differ significantly from LPS alone. NBF (concentration: 20 µg/mL) counteracted this dramatic increase and suppressed the release to the naïve/vehicle level range. NBF alone did not differ from the control groups i.e., no effect by itself (Figure 5).

### 3.6. The Effect of LPS with or without NBF on the Gene Expression of NF-κB, TLR2 and TLR4

NF-κB, TLR2, and TLR4 mRNA levels were analyzed following exposure of the cells to LPS with or without NBF as compared to control.

The vehicle group showed the lowest mRNA levels of NF-κB, TLR2, and TLR4, while these levels were increased 3 h after exposure to LPS, by 27%, 29% and 22%, respectively, which shows an induction of inflammatory response after exposure to LPS. NBF treatment significantly elevated the TLR2 and TLR4 mRNA expression compared to the LPS group (48% and 26%, respectively), without a significant effect on NF-κB (Figure 6). Primer sequences are listed in the Appendix A.

## 4. Discussion

LPS is known for inducing inflammatory responses in macrophages [37]. The present study investigated the in vitro anti-inflammatory effects of NBF in RAW264.7 macrophages stimulated by LPS. The basic constituents of NBF, namely, botanical cannabidiol and ME, have been demonstrated to independently possess anti-inflammatory and immunomodulatory effects [29,38]. Each of the five MMs from NBF together were expected to synergistically or cumulatively enhance the effects on the immune system by activating different immune pathways in the host including incising macrophages, natural killer cells, monocytes, neutrophil, T-cells, B-cells, dendritic cells, and chemical mediators (for example, cytokines). In the present study, we investigated the anti-inflammatory effect of NBF which contained a combination of constituents in an in vitro model of inflammation using the RAW264.7 murine macrophage cell line stimulated with LPS.

In order to assess the effect of NBF on LPS-induced RAW264.7 cells, several inflammatory markers were analyzed, including IL-1β, TNF-α, IL-6, CCL5, COX-2, NO, and LDH (as a marker for necrosis). As we hypothesized, NBF was able to completely counteract the LPS-induced release of IL-6, and partially counteract the release of IL-1β and TNF-α (Figure 2). TNF-α is a potent cytokine which activates inflammatory pathways and can cause several inflammatory diseases. Excessive production of TNF-α by macrophages can also lead to serious pathological conditions such as septic shock and rheumatoid arthritis [39]. IL-1β is a cytokine which belongs to the IL-1 peptide group. IL-1β is suggested to play a role in tumorigenesis as well as inflammatory activation [40]. CBD directly interferes with TLR 2 and 4 signal transduction by activating the NF-κB pathway [41]. The results suggest inhibitory effects of NF-κB that are associated with NF-κB p65.

NO is a very unstable and harmful molecule which is the product of arginine metabolism and is released during the activation of the M1 inflammatory pathway. This unstable molecule is converted into a stable molecule nitrite/nitrate [42]. NBF inhibited the production of NO in LPS-stimulated RAW264.7 cells to control range, indicating the complete inactivation of reactive nitrogen species production. Moreover, NBF also showed the complete inhibition of COX-2 elevation in LPS-induced RAW264.7 cells. Prostaglandins are lipid molecules, and they are formed from arachidonic acid [43]. They mediate pathogenic mechanisms and are important markers of inflammatory response [44]. Such inhibition of COX-2 expression makes NBF applicable for the treatment of inflammatory diseases [45].

CCL5 is a chemokine which binds to the CCR5 receptor and other chemokine receptors [46]. CCL5 is known to be an important marker in inflammation and is overexpressed in inflammatory states [47]. NBF completely inhibited the CCL5 production in LPS-stimulated RAW264.7 cells, which shows that this formulation does have immunomodulatory effects, as reflected by suppression of CCL5 expression, which is responsible for recruiting leukocytes at a site of inflammation [48].

NBF did not induce LDH secretion from the RAW264.7 cells, an observation that verified the safety of NBF (Figure 1), at least under in vitro conditions. The anti-inflammatory activity of the NBF may be related to an interaction with scavenger receptors (SRs), primarily SR-E2 (CD369) and SR-F1 (SCARF1) which, along with tetraspanins, form complex clusters with TLR and β-glucan on the cell membrane and affect TLR activity [49,50]. Additionally, an interaction with pattern recognition receptors (PRRs) [50] may also be relevant to the immunomodulatory effects of the NBF. Moreover, in the present study, we did not assess markers of M2 macrophages, such as IL-10 and TGF-β. Thus, the possibility of polarization of M1 to M2 macrophages following treatment with NBF is unclear at present. As shown in Appendix A, XTT analysis was performed following exposure of the cells to (ME) and to LPS. The results indicated that LPS significantly decreased cell viability, however, ME has no effect. 

Previous studies have also demonstrated the in vitro immunomodulatory effects of some ME on LPS-stimulated RAW264.7 macrophage cells [51,52]. CBD alone was also found to possess immunomodulatory effects in same cells. However, in our study, we assessed the impact of a combination of ME and CBD and found that the combination (NB) achieved a significant general reduction in all the tested inflammatory markers, a phenomenon that was not achieved in the studies of each of them alone. Thus, it seems that the current formulation of NBF is a promising candidate in the field of plant-derived anti-inflammatory agents. In the future, we intend to compare the immunomodulatory activity of the ME and CBD separately.

## 5. Conclusions

This study shows that NBFs possess anti-inflammatory and immunomodulatory effects, which not only inhibit the release of pro-inflammatory cytokines, but also the production of relevant chemokine (CCL5), prostaglandins (inhibition of COX-2), and nitrite release. This potent and broad anti-inflammatory activity may be relevant to several inflammatory diseases. Although NBF did not display a cytotoxic effect on RAW274.7 cells, as reflected by the lack of LDH elevation, in vitro studies with non-immune cells and in vivo studies are needed to ascertain its clinical use and safety. The mRNA analyses also demonstrated a stimulatory effect of NBF on NF-κB, the expression of TLR2 and TLR4. These increases, without an impact on NF-κB expression, may represent a compensatory mechanism geared to adapt to the suppression of proinflammatory cytokine expression. Although the results are encouraging, the relevance of TLR gene expression to the therapeutic role of NBF in inflammatory diseases merits further investigation of in vitro and in vivo models of inflammation and assessment of gene and protein expression profiles and enriched pathways.

## Figures and Tables

**Figure 1 jof-08-00321-f001:**
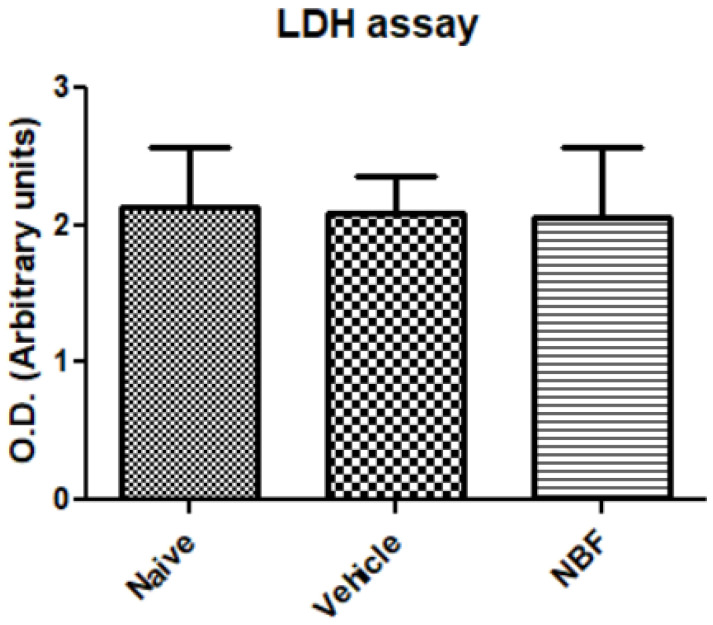
Cytotoxicity analysis as assessed by LDH release from RAW264.7 macrophages. LDH levels were measured by O.D. (arbitrary units) and are presented as the mean ± SD. At least 8 replicates were in each group (*n* = 8 for naïve, *n* = 16 for vehicle, and *n* = 14 for NBF). ANOVA with Bonferroni’s post hoc test was performed. *p*-value = 0.92.

**Figure 2 jof-08-00321-f002:**
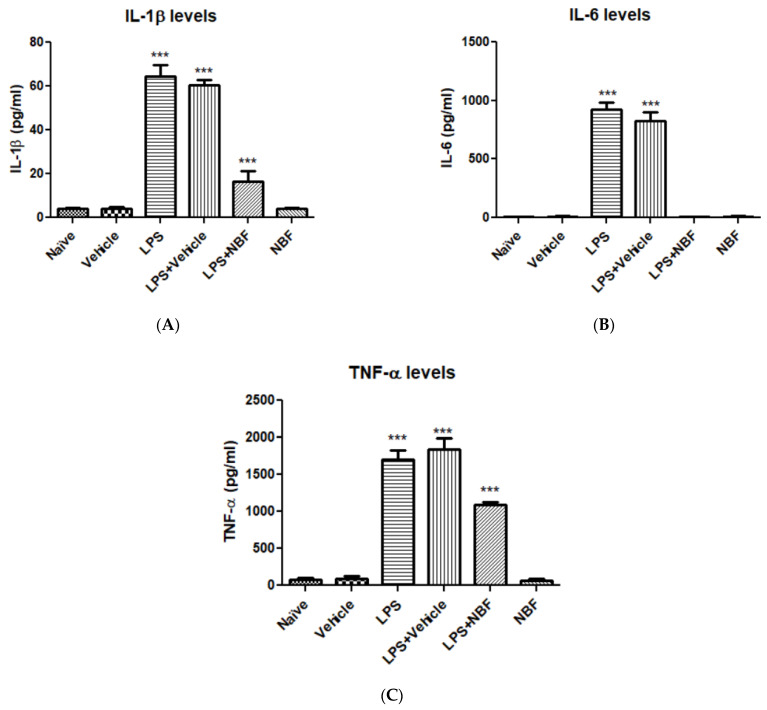
NBF downregulates the release of pro-inflammatory cytokines induced by LPS in RAW264.7 macrophages. RAW 264.7 macrophages were exposed to 20 ng/mL LPS and simultaneously with the NBF, 20 µg/mL. IL-1β (**A**), IL-6 (**B**), and TNF-α (**C**) levels were measured using ELISA. The results are presented as the mean ± SD, (*n* = 4 in each group). ANOVA followed by Bonferroni’s post hoc test was performed. *** *p* < 0.001 compared to all other groups.

**Figure 3 jof-08-00321-f003:**
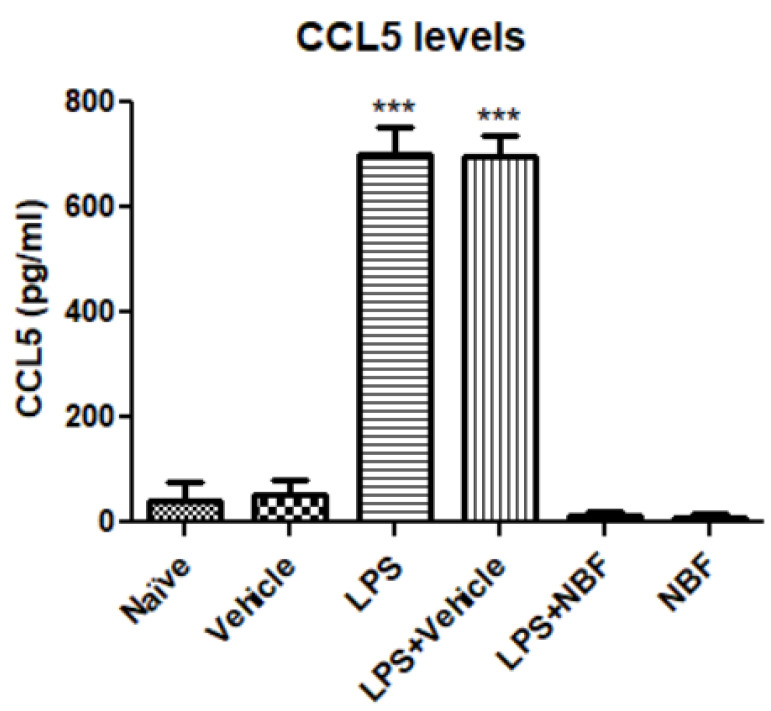
NBF downregulates the release of CCL5 (RANTES) in RAW264.7 macrophages following LPS stimulation. RAW 264.7 macrophages were exposed to 20 ng/mL LPS and simultaneously with the NBF, 20 µg/mL. CCL5 levels were measured (pg/mL) using standard calibration curves and are presented as the mean ± SD. Four replicates in each group (*n* = 4). ANOVA followed by Bonferroni’s post hoc test was performed. *** *p* < 0.001 compared to all other groups.

**Figure 4 jof-08-00321-f004:**
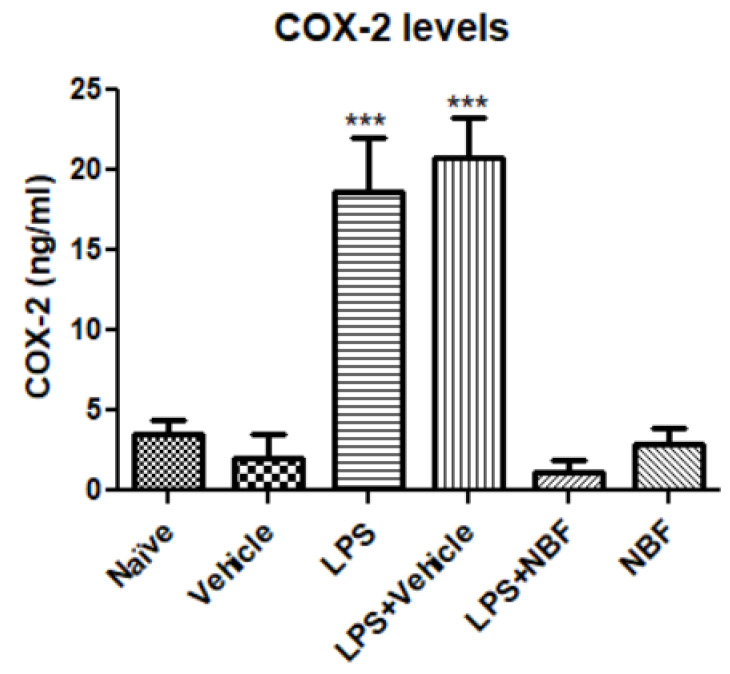
NBF affects COX-2 protein level in RAW264.7 macrophages following LPS stimulation. RAW 264.7 macrophages were exposed to 20 ng/mL LPS and simultaneously with the NBF, 20 µg/mL. COX-2 levels were measured by ELISA using standard calibration curve and are presented as the mean ± SD. Four replicates in each group (*n* = 4). ANOVA followed by Bonferroni’s post hoc test was performed. *** *p* < 0.001 compared to all other groups.

**Figure 5 jof-08-00321-f005:**
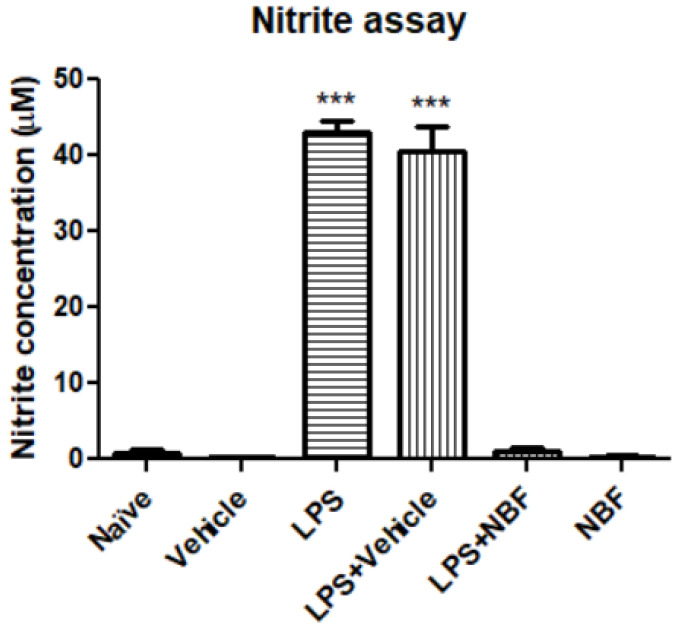
NBF downregulates the secretion of nitrite in RAW264.7 macrophages following LPS stimulation. RAW 264.7 macrophages were exposed to 20 ng/mL LPS and simultaneously with the NBF, 20 µg/mL. Nitrite levels were measured using standard calibration curve with sodium nitrite and are presented as the mean ± SD. Four replicates in each group (*n* = 4). ANOVA followed by Bonferroni’s post hoc test was performed. *** *p* < 0.001 compared to all other groups.

**Figure 6 jof-08-00321-f006:**
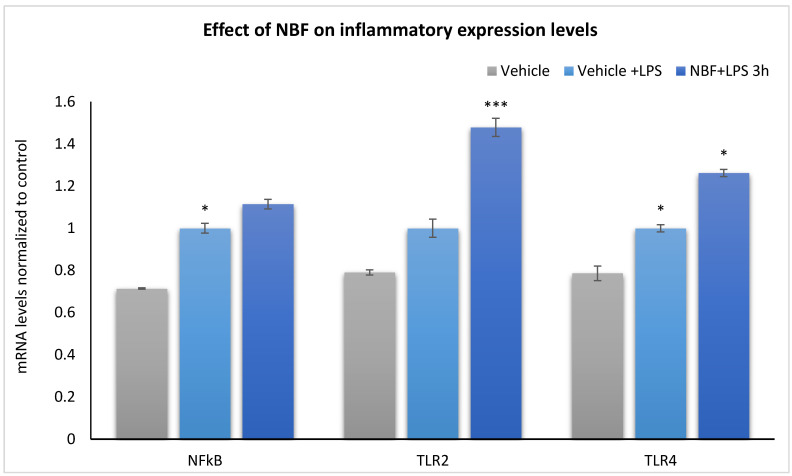
NBF upregulates mRNA expression levels of TLR2 and TLR4 but not NF-κB. RAW264.7 cells were treated with either Vehicle + LPS or NBF + LPS for 3 h. mRNA was extracted 3 h after treatment using an mRNA extraction kit and NF-κB, TLR2, and TLR4 mRNA were quantified and normalized to the glyceraldehyde 3-phosphate dehydrogenase (GAPDH)—mRNA expression levels are shown as the fold differences compared to untreated cells. Data are represented as the mean  ±  SEM of three independent experiments (three separate experiments in triplicates) and statistical analyses were performed using ANOVA followed by Bonferroni’s post hoc test. * *p* < 0.05, *** *p* < 0.001 vs. vehicle.

**Table 1 jof-08-00321-t001:** The components in mushroom extracts (ME).

#	Mushroom Extract	Extraction Method	% Polysaccharides	% Triterpenes	Heavy Metals, Hg	Heavy Metals Pb, As
1	*Cordyceps militaris*	Hot water	33.2%	-	≤0.2 mg/kg	≤1 mg/kg
2	*Flammulina velutipes*	Hot water	32.8%	-	≤0.2 mg/kg	≤1 mg/kg
3	*Ganoderma lucidum*	Hot water	32.7%	-	≤0.2 mg/kg	≤1 mg/kg
4	*Ganoderma lucidum*	Ethanol	32.7%	6.3%	≤0.2 mg/kg	≤1 mg/kg
5	*Lentinula edodes*	Hot water	32.0%	-	≤0.2 mg/kg	≤1 mg/kg
6	*Trametes versicolor*	Hot water	78%	-	≤0.2 mg/kg	≤1 mg/kg

## Data Availability

All the data are available but our permission should be requested for their use.

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
