# Peer review of "The Effect of Natural-Based Formulation (NBF) on the Response of RAW264.7 Macrophages to LPS as an In Vitro Model of Inflammation"

_jof, 2022, doi:10.3390/jof8030321_

Round 1

Reviewer 1 Report

Dear Authors,
In general, positively evaluating your work, I have to make a number of comments and ask you a few questions.

(96-97) Reference [33] is mainly devoted to the antioxidant properties of mushroom extracts. However, this is clearly insufficient to justify the effect of such drugs on macrophages. In addition to TLR and CR, fungal antigens, for example, can bind to scavenger receptors (SRs), primarily SR-E2 (CD369) and SR-F1 (SCARF1). At the same time, SR, along with tetraspanins, form complex clusters with TLR on the cell membrane and thereby regulate TLR activity. Perhaps you should have elaborated here and in the Discussion section on these and other mechanisms of PRR regulation by fungal antigens.

(119) RAW264.7 cells were treated with 20 μg/ml of “Immunity formulation”. However, in the article hereinafter there is no justification for the use of this particular dose of “Immunity”.

(128-143) In the study, M1 markers (IL-1β, TNF-α, CCL5, NO, NF-κB), IL-6 (M1/M2 marker) were predominantly determined, but not classical M2 markers (IL-10 and TGFβ). The authors did not set themselves the task of investigating the possibility of polarization of macrophages under the action of the drug in the direction of M2?

(156) A dose of 20 ng/ml of LPS was used. However, this is a small dose. Commonly used in vitro doses are 20-500 ng/ml of LPS. It is not clear why the dose of 20 ng/ml was chosen and why the dose-response principle was not used, for example, two doses - 20 and 200 ng/ml? Perhaps then the results would not be so unambiguous.

(181-186) The authors used statistics designed only for the normal distribution of data within groups (mean ± standard deviation, Student's t-test). However, the authors did not indicate whether there was a normal distribution of data in the study groups.

(271) LPS is known for inducing inflammatory responses in microglia and macrophages. I don't understand what microglia has to do with it? Microglia are stromal macrophages of the brain.

Author Response

(96-97) Reference [33] is mainly devoted to the antioxidant properties of mushroom extracts. However, this is clearly insufficient to justify the effect of such drugs on macrophages. In addition to TLR and CR, fungal antigens, for example, can bind to scavenger receptors (SRs), primarily SR-E2 (CD369) and SR-F1 (SCARF1). At the same time, SR, along with tetraspanins, form complex clusters with TLR on the cell membrane and thereby regulate TLR activity. Perhaps you should have elaborated here and in the Discussion section on these and other mechanisms of PRR regulation by fungal antigens.

Response: This concern was addressed in the discussion from line 312-317.

(119) RAW264.7 cells were treated with 20 μg/ml of “Immunity formulation”. However, in the article hereinafter there is no justification for the use of this particular dose of “Immunity”.

Response: Supplementary figure 1 of dose response analysis was added and was referred to the methods.

(128-143) In the study, M1 markers (IL-1β, TNF-α, CCL5, NO, NF-κB), IL-6 (M1/M2 marker) were predominantly determined, but not classical M2 markers (IL-10 and TGFβ). The authors did not set themselves the task of investigating the possibility of polarization of macrophages under the action of the drug in the direction of M2?

Response: In this study, we focused only on the suppression of M1 inflammatory markers. In the future, we will investigate the possibility of macrophage polarization.

(156) A dose of 20 ng/ml of LPS was used. However, this is a small dose. Commonly used in vitro doses are 20-500 ng/ml of LPS. It is not clear why the dose of 20 ng/ml was chosen and why the dose-response principle was not used, for example, two doses - 20 and 200 ng/ml? Perhaps then the results would not be so unambiguous.

Response: Usually, immune cells like macrophages are very responsive to LPS. Macrophages are particularly considered to be the most sensitive and responsive to TLR ligands, such as LPS, IFN-γ or both. In the future, we will attempt to use a concentration of 200 ng/ml as a comparator.

(181-186) The authors used statistics designed only for the normal distribution of data within groups (mean ± standard deviation, Student's t-test). However, the authors did not indicate whether there was a normal distribution of data in the study groups.

Response: In all experiments, there was a normal distribution of data in the study groups. We used Prism software for statistical analyses.

(271) LPS is known for inducing inflammatory responses in microglia and macrophages. I don't understand what microglia has to do with it? Microglia are stromal macrophages of the brain.

Response: The microglia was deleted and references were edited to just macrophages.

Reviewer 2 Report

Review report

Journal: Journal of Fungi

Manuscript ID: jof-1618618

Type: Article

Title: The effect of Immunity formulation on the response of RAW264.7 macrophages to LPS as an in vitro model of inflammation

Acute or chronic inflammation is a very complex process involving many different cells, cytokines, transcription factors, etc. Currently used anti-inflammatory drugs are effective, but they have many adverse effects, especially when given for a long time. The better knowledge about the inflammation pathways gives the possibility to interact with the new compounds of natural origin or synthetic. Plants, fruits, or fungi are the reach source of different known and unknown substances with great therapeutic potential. Despite the essence of the problem raised in the study, I have some major and minor remarks, listed below. I hope that may help to improve the manuscript.

Introduction part

  • please consider moving the part (lines 53-84) at the beginning of the Discussion and explain more about the action of CBD on the immune system; it is not clear why did Authors decide to combine medicinal mushroom extracts with CBD

Materials and Methods

  • please provide the exact method of obtaining the extract from medicinal mushrooms and CBD formulation (if it was commercially purchased, please give the cat. number); were the active substances verified/analyzed using e.g., HPLC or other methods
  • please add the origin of the RAW264.7 macrophage cell line and manufacturers for all used chemicals and reagents (not only ELISA or rtPCR)
  • what exactly was compared with Student’s t-test? please explain and move it to the subsection with statistical analysis

Discussion

  • please consider explaining here the mechanisms of action of medicinal mushrooms extract and CBD
  • make a conclusion as a separate part

References need careful revision; some examples are given below.

  • ref. 1, 6, 8, 12, 19, 22, 27, 28, and 29 – correct name of the Authors (provide full family names, only abbreviations are given)
  • ref. 13, 32, 42, and 44 – check for DOI number
  • ref. 33 – please provide journal information (name, volume, pages, DOI)
  • ref. 41 – please correct – there is no reference given

Minor remarks

  • correct typing mistakes e.g., lines 51, 64, 70, 84, 90, 96, 337

Revise and prepare the manuscript according to the requirements of the Publisher. Some examples are listed below.

  • specify the kind of support given by all companies – without this it is not possible to exclude the conflict of interests
  • add abbreviations of Authors’ names and email addresses on the title page
  • correct “Authors Contribution” part using Authors’ names abbreviations and correct contribution due to the statements mentioned in the template of Journal of Fungi
  • add “Data Availability Statement” and “Conflicts of Interest” statements

Author Response

Acute or chronic inflammation is a very complex process involving many different cells, cytokines, transcription factors, etc. Currently used anti-inflammatory drugs are effective, but they have many adverse effects, especially when given for a long time. The better knowledge about the inflammation pathways gives the possibility to interact with the new compounds of natural origin or synthetic. Plants, fruits, or fungi are the reach source of different known and unknown substances with great therapeutic potential. Despite the essence of the problem raised in the study, I have some major and minor remarks, listed below. I hope that may help to improve the manuscript.

Introduction part

  • please consider moving the part (lines 53-84) at the beginning of the Discussion and explain more about the action of CBD on the immune system; it is not clear why did Authors decide to combine medicinal mushroom extracts with CBD
  • Response: CBD was combined with medicinal mushroom extracts since it also possesses immunomodulatory activity.

Materials and Methods

  • please provide the exact method of obtaining the extract from medicinal mushrooms and CBD formulation (if it was commercially purchased, please give the cat. number); were the active substances verified/analyzed using e.g., HPLC or other methods
  • Response: The synthesis was added from line 116-130
  • please add the origin of the RAW264.7 macrophage cell line and manufacturers for all used chemicals and reagents (not only ELISA or rtPCR)
  • Response: The issue was addressed in line 107-108, 111, and 115.
  • what exactly was compared with Student’s t-test? please explain and move it to the subsection with statistical analysis
  • Response: The results were compared to the control group (LPS alone) line 194-195.

Discussion

  • please consider explaining here the mechanisms of action of medicinal mushrooms extract and CBD.
  • Response: We added several mechanisms that may be related the immunomodulatory activity of Immunity. The issue was mentioned in the discussion from line 319-327.
  • make a conclusion as a separate part
  • Response: The part was separated in line 328

References need careful revision; some examples are given below.

  • ref. 1, 6, 8, 12, 19, 22, 27, 28, and 29 – correct name of the Authors (provide full family names, only abbreviations are given)
  • ref. 13, 32, 42, and 44 – check for DOI number
  • ref. 33 – please provide journal information (name, volume, pages, DOI)
  • ref. 41 – please correct – there is no reference given
  • Response: References were fixed

Minor remarks

  • correct typing mistakes e.g., lines 51, 64, 70, 84, 90, 96, 337
  • Response: typing mistakes were fixed

Revise and prepare the manuscript according to the requirements of the Publisher. Some examples are listed below.

  • specify the kind of support given by all companies – without this it is not possible to exclude the conflict of interests
  • Response: The Cannabotech company paid the salary of Dr. Sheelu Monga and sponsored the lab expenses for this project.
  • add abbreviations of Authors’ names and email addresses on the title page
  • Response: abbreviations of authors names and email addresses were added on title page
  • correct “Authors Contribution” part using Authors’ names abbreviations and correct contribution due to the statements mentioned in the template of Journal of Fungi
  • Response: It was corrected according to the guidelines
  • add “Data Availability Statement” and “Conflicts of Interest” statements
  • Response: These two statements were added in the end of the manuscript.

Reviewer 3 Report

The study by Sheelu Monga et al entitled “The effect of Immunity formulation on the response of RAW264.7 macrophages to LPS as an in vitro model of inflammation” has been reviewed. The study showed the anti-inflammatory efficacy of a commercial formulation called Immunity (mushroom-cannabidiol extract) on RAW264.7 macrophages exposed to lipopolysaccharide. The manuscript is quite well written even if approximate in some sections and also not correctly designed. I think the authors should make a more concerted effort to add, in a rigorous and detailed way, all useful information so that all reported results are reproducible and reliable.

Please revise the points described below:

- Please avoid the use of the formulation commercial name throughout the manuscript including the title.

-Use the binomial nomenclature for all listed species.

-In the introduction, please describe the biological activities reported in the scientific literature concerning each of the reported mushroom species and also for cannabidiol.

-Lane 85 the references are not relevant to the period reported.

-Lane 97 the reference is incorrect.

-In materials and methods, not enough information has been provided on the nature of the different medicinal mushroom and cannabis extracts, please report:

1-Origin and authentication of all raw materials.

2-Extraction method for each extract including the yield.

3-The characterization of the single extract by reporting the quantity of the major phytochemicals present in the formulation.

4- Please report the list of ingredients present in the final preparation including the nominal quantities of extract that compose it.

5- Add further product specifications such as parameters on purity etc.

-In materials and methods please report the origin of cell culture and the exact protocol of treatment for the formulation (time of exposure, pre- or post-treatment with lps etc.

-Cell cytotoxicity and proliferation assays are generally used for drug screening to detect whether the test molecules have effects on cell proliferation or display direct cytotoxic effects. Why did the authors not plan to also test cell viability to establish the correct concentration to use in the study and the possible toxicity of the higher doses? The single-dose LDH test, however, is not sufficient to ascertain the safety of the formulation.

-How did the authors select the concentration used in this study?

- Please report all useful information about the kits and products used in your research, such information is missing in some paragraphs (ELISA).

- No description is present about the vehicle used.

- The discussion is very weak and only the results are reported together with the description of the markers analysed. Please review the entire section by discussing the properties of the formulation studied against the available literature data and highlighting

the best properties of the formulation compared to single extracts if present.

-The manuscript requires significant editorial correction.

-Please revise the references list according to journal guidelines

Author Response

The study by Sheelu Monga et al entitled “The effect of Immunity formulation on the response of RAW264.7 macrophages to LPS as an in vitro model of inflammation” has been reviewed. The study showed the anti-inflammatory efficacy of a commercial formulation called Immunity (mushroom-cannabidiol extract) on RAW264.7 macrophages exposed to lipopolysaccharide. The manuscript is quite well written even if approximate in some sections and also not correctly designed. I think the authors should make a more concerted effort to add, in a rigorous and detailed way, all useful information so that all reported results are reproducible and reliable.

Please revise the points described below:

- Please avoid the use of the formulation commercial name throughout the manuscript including the title.

Response: The NBF term was used instead of Immunity formulation

-Use the binomial nomenclature for all listed species.

Response: Table 1 was added for binomial nomenclature.

-In the introduction, please describe the biological activities reported in the scientific literature concerning each of the reported mushroom species and also for cannabidiol.

Response: The biological activities of each mushroom and CBD was added.

-Lane 85 the references are not relevant to the period reported.

Response: The incorrect reference was removed

-Lane 97 the reference is incorrect.

Response: The reference was changed

-In materials and methods, not enough information has been provided on the nature of the different medicinal mushroom and cannabis extracts, please report:

1-Origin and authentication of all raw materials.

Response: The materials origin and it’s details were added

2-Extraction method for each extract including the yield.

Response: Table 1 was added with details.

3-The characterization of the single extract by reporting the quantity of the major phytochemicals present in the formulation.

Response: The details of each single mushroom characteristic was added in the introduction

4- Please report the list of ingredients present in the final preparation including the nominal quantities of extract that compose it.

Response: The details were added in Table 1

5- Add further product specifications such as parameters on purity etc.

-In materials and methods please report the origin of cell culture and the exact protocol of treatment for the formulation (time of exposure, pre- or post-treatment with lps etc.

Response: The origin of the cells was added and it is a simultaneous treatment which was given to the cells for 24 hours (mentioned in methods)

-Cell cytotoxicity and proliferation assays are generally used for drug screening to detect whether the test molecules have effects on cell proliferation or display direct cytotoxic effects. Why did the authors not plan to also test cell viability to establish the correct concentration to use in the study and the possible toxicity of the higher doses? The single-dose LDH test, however, is not sufficient to ascertain the safety of the formulation.

Response: Supplementary figure 2 was added for the same.

-How did the authors select the concentration used in this study?

Response: Supplementary figure 1 was added for the same.

- Please report all useful information about the kits and products used in your research, such information is missing in some paragraphs (ELISA).

Response: Missing information was added in line 168.

- No description is present about the vehicle used.

Response: Vehicle information was added in line 133-134.

- The discussion is very weak and only the results are reported together with the description of the markers analyzed. Please review the entire section by discussing the properties of the formulation studied against the available literature data and highlighting

Response: The discussion was improved and relates to previous studies, with the same cellular model, in this field

the best properties of the formulation compared to single extracts if present.

Response: Unfortunately, we don’t have data on single extracts

-The manuscript requires significant editorial correction.

Response: The style of the manuscript was improved.

-Please revise the references list according to journal guidelines

Response: The references were revised according to the journal guidelines.

Round 2

Reviewer 3 Report

The authors satisfied most of the concerns highlighted, but there are still some unanswered requests which are reported below.

 -Please avoid the use of the formulation commercial name throughout the manuscript including the title.

The trade name still appears in the title

2- The extraction method for each extract including the yield,

It has not yet been clarified by the authors. For scientific rigour, every aspect of the study must be reproducible and reliable. 

The list of ingredients present in the final preparation including the nominal quantities of each extract that compose the natural-based formulation ( mg/g or %) is missing!!!

I would urge the authors to accommodate the suggestions reported above, to further improve the quality of the manuscript.

Author Response

We Thank the reviewers for their constructive comments and suggestions. 

 -Please avoid the use of the formulation commercial name throughout the manuscript including the title.

Response: The formulation commercial name was removed throughout the manuscript, including title.

The trade name still appears in the title

2- The extraction method for each extract including the yield,

It has not yet been clarified by the authors. For scientific rigour, every aspect of the study must be reproducible and reliable. 

The list of ingredients present in the final preparation including the nominal quantities of each extract that compose the natural-based formulation ( mg/g or %) is missing!!!

Response: The methods section was entirely modified, including the addition of minor details.